# Supporting Aboriginal Community Controlled Health Services to deliver alcohol care: protocol for a cluster randomised controlled trial

Kristie H Harrison,[1] KS Kylie Lee,[2,3] Timothy Dobbins,[4] Scott Wilson,[2,5] Noel Hayman,[6,7] Rowena Ivers,[1,8] Paul S Haber,[2,9] James H Conigrave,[2] David Johnson,[10] Beth Hummerston,[10] Dennis Gray,[11] Katherine Conigrave [2,9]

**Correspondence to**
Professor Katherine Conigrave;
kate.conigrave@sydney.edu.au

## ABSTRACT

**Introduction** Indigenous peoples who have experienced colonisation or oppression can have a higher prevalence of alcohol-related harms. In Australia, Aboriginal Community Controlled Health Services (ACCHSs) offer culturally accessible care to Aboriginal and Torres Strait Islander (Indigenous) peoples. However there are many competing health, socioeconomic and cultural client needs.

**Methods and analysis** A randomised cluster wait-control trial will test the effectiveness of a model of tailored and collaborative support for ACCHSs in increasing use of alcohol screening (with Alcohol Use Disorders Identification Test-Consumption (AUDIT-C)) and of treatment provision (brief intervention, counselling or relapse prevention medicines).

**Setting** Twenty-two ACCHSs across Australia.

**Randomisation** Services will be stratified by remoteness, then randomised into two groups. Half receive support soon after the trial starts (intervention or 'early support'); half receive support 2 years later (wait-control or 'late support').

**The support** Core support elements will be tailored to local needs and include: support to nominate two staff as champions for increasing alcohol care; a national training workshop and bimonthly teleconferences for service champions to share knowledge; onsite training, and bimonthly feedback on routinely collected data on screening and treatment provision.

**Outcomes and analysis** Primary outcome is use of screening using AUDIT-C as routinely recorded on practice software. Secondary outcomes are recording of brief intervention, counselling, relapse prevention medicines; and blood pressure, gamma glutamyltransferase and HbA1c. Multi-level logistic regression will be used to test the effectiveness of support.

**Ethics and dissemination** Ethical approval has been obtained from eight ethics committees: the Aboriginal Health and Medical Research Council of New South Wales (1217/16); Central Australian Human Research Ethics Committee (CA-17-2842); Northern Territory Department of Health and Menzies School of Health Research (2017-2737); Central Queensland Hospital and Health Service (17/QCQ/9); Far North Queensland (17/QCH/45-1143); Aboriginal Health Research Ethics Committee, South Australia (04-16-694); St Vincent's Hospital (Melbourne)

## Strengths and limitations of this study

► This large cluster randomised controlled trial provides the power to test whether a model of support for Aboriginal and Torres Strait Islander primary care services can increase rates of alcohol screening and treatment provision.

► The protocol has been designed to be compatible with cultural context and to integrate western knowledge with the expertise and holistic approaches of these services.

► The use of regular data feedback and nomination of service champions offers the services an opportunity to be involved in ongoing quality improvement on alcohol care.

► The resultant study will be able to use routinely collected outcome data but this relies on the accurate recording of screening and alcohol care provided to clients.

Human Research Ethics Committee (LRR 036/17); and Western Australian Aboriginal Health Ethics Committee (779).

**Trial registration number** ACTRN12618001892202; Pre-results.

## INTRODUCTION

Globally, alcohol is the leading cause of loss of healthy life years for people between the ages of 15 and 49.[1] Indigenous peoples dealing with colonisation and oppression can be at higher risk of mental illness and harms from alcohol.[2] In Australia, the proportion of Aboriginal and Torres Strait Islander (Indigenous) people who drink alcohol is less than non-Indigenous people.[3] However, Indigenous Australians are 2–8 times more likely to be hospitalised for alcohol-related conditions[4] and nine times more likely to die from alcohol-related harms.[5] This directly impacts on the health gap between Indigenous and

non-Indigenous Australians.[4] These increased risks and harms from drinking can be related to experiences of trauma, grief, poverty and cultural dispossession, resulting in increased mental illness, trans-generational trauma and lower life expectancy.[2 4 6] The lack of community autonomy and access to culturally appropriate treatment services also contribute.[4]

Early detection of unhealthy drinking (ie, of hazardous, harmful or dependent alcohol use)[7 8] plays a key role in reducing the overall burden of alcohol conditions, particularly before people develop health problems or social impacts such as incarceration related to alcohol use.[9–11] Internationally, alcohol screening and brief intervention approaches have been found to be cost-effective in reducing unhealthy drinking, particularly for those who are not dependent on alcohol.[9 12] However across Australian primary care settings, screening and brief intervention for unhealthy drinking are not always standardised,[13] informed by Australian National Health and Medical Research Council (NHMRC) drinking guidelines,[14] or systematically conducted. It is estimated that 50%–70% of people with unhealthy drinking go undetected in Australian primary care.[15] Similarly only 57% of attendees were screened for unhealthy drinking in four Indigenous primary care settings in Queensland, Australia (in 2007).[16] Also, when alcohol dependence is present, pharmacological treatments to reduce risk of relapse are not regularly prescribed in primary healthcare.[17–19]

Culturally specific healthcare services that are led and delivered by Indigenous peoples can have a key role in improving quality of healthcare and treatment access.[20] Aboriginal Community Controlled Health Services (ACCHSs) provide accessible and culturally appropriate healthcare to Indigenous communities across Australia. Each ACCHS has its own holistic service delivery model for the communities that they serve.[20] As with mainstream (general population) services, knowledge and practice of screening and brief intervention vary between and within Indigenous primary care services.[13 21 22] Some ACCHSs have an alcohol and other drug (AOD) worker or team, or links with external AOD specialists/services. Others may have no specific AOD expertise and health practitioners may lack confidence in delivering alcohol care (ie, screening, brief intervention, counselling or medicines). Prior to 2017, the Australian government had asked ACCHS staff to record clients' alcohol use as 'safe' or 'unsafe', 'non-drinker' or 'ex-drinker'.[22] However health practitioners may or may not have known current drinking guidelines or have applied these to assess the risk.

To increase standardisation of alcohol screening in ACCHSs, in July 2017 the use of the Alcohol Use Disorders Identification Test-Consumption (AUDIT-C) was added as a national key performance indicator for ACCHSs.[23] AUDIT-C[24] is a 3-item screening tool, comprised of the first three questions of the 10-item AUDIT.[25] This short form has been shown to have good sensitivity and specificity in comparison to the full AUDIT.[24] Several studies suggest that AUDIT-C is valid in Indigenous primary care settings, though further study into culturally appropriate delivery is suggested.[26–28]

Internationally, research suggests that support can help primary care services increase implementation of alcohol screening and brief intervention[29] or provision of pharmacotherapies for relapse prevention in alcohol dependence.[19] Support which is multi-faceted, and targets several areas of a health service seems particularly promising.[29] For example the approach may target not just doctors, but other professionals, clients, and organisational systems.[29] Continuing quality improvement approaches may offer sustainable benefit, including in ACCHSs.[30] However there have been very few studies of culturally relevant support models for ACCHSs in alcohol care,[21 30] with even fewer conducted by Indigenous researchers.

This paper outlines the trial protocol for testing if a model of support can assist ACCHSs to integrate the mainstream evidence base on alcohol care with their own cultural and clinical expertise. The use of an inactive control group can be challenging on ethical and partnership grounds, particularly for vulnerable communities. In contrast, a wait-control design allows support to be delivered to all participating ACCHSs.

## AIMS

This trial aims to assess if a model of tailored and collaborative support for ACCHSs can result in increased uptake of evidence-based screening and treatment for unhealthy alcohol use (ie, drinking above recommended limits). This service-wide model of support will be designed to build on services' existing strengths (figure 1).

## METHODS
### Overview

A cluster randomised trial will test the effectiveness of a model of tailored and collaborative support to increase the use of AUDIT-C and evidence-based treatment in ACCHSs compared with care as usual. Culturally relevant supports will be offered with the aim of increasing staff

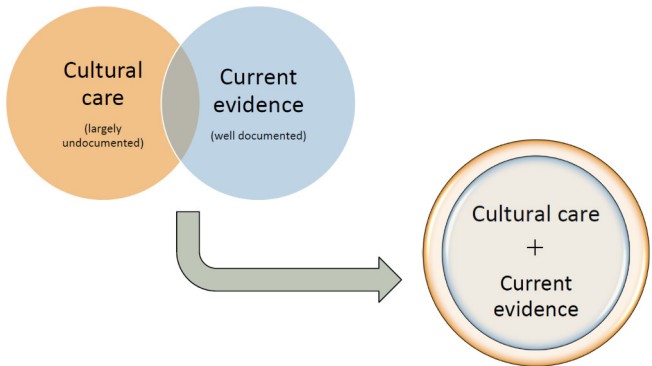

**Figure 1** Project principles—embedding mainstream evidence into a cultural care approach.

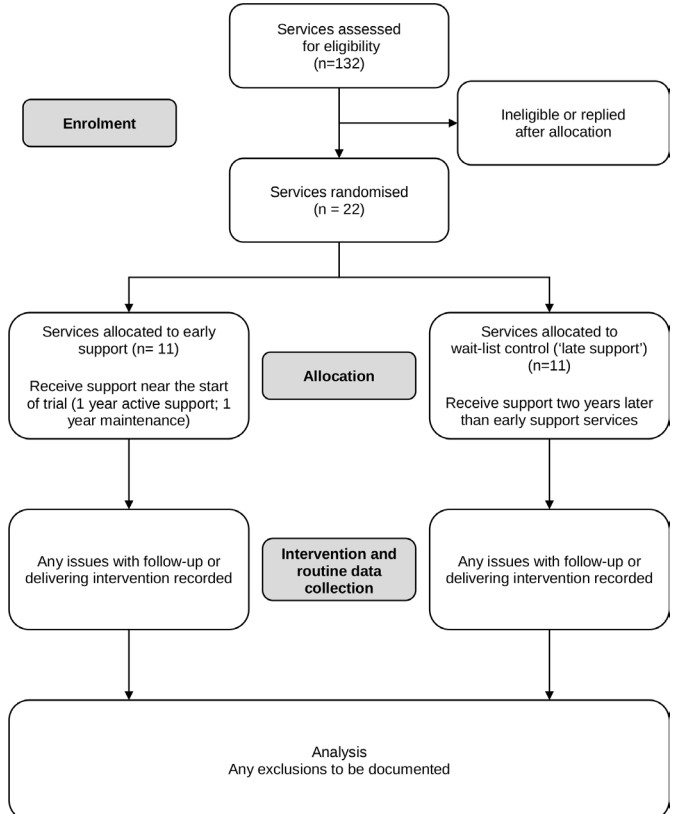

**Figure 2** Flow diagram for the trial.

The support provided will contain eight core elements (table 1), but be tailored, guided by ACCHSs, to their individual needs, including needs by remoteness. It will also be underpinned by current evidence such as national alcohol treatment guidelines,[10] Alcohol Treatment Guidelines for Indigenous Australians[31] and national alcohol consumption guidelines.[14] The study methods will be based on principles of conducting health research with Indigenous Australians.[32] This includes Indigenous involvement and engagement, benefits for Indigenous peoples, and led by Indigenous Australians.

The primary outcome will be routinely collected data on screening rates using AUDIT-C. Secondary outcomes will include recorded delivery of brief intervention, counselling or prescribed medicines to reduce relapse in alcohol dependence. We will also describe changes in reported drinking and in biological measures that can be affected by drinking such as gamma glutamyltransferase (GGT, a liver enzyme),[33] HbA1C[34] and systolic blood pressure (BP).[34]

### Participating services and eligibility
#### Recruitment
This trial will recruit whole services (no clients will be recruited). ACCHSs across Australia will be recruited based on expression of interest after circulation of study information to eligible services. A mix of services will be approached (by KHH, BH or KC) with a range of remoteness as determined by the Australian Standard Geographical Classification—Remoteness Area (ASGC-RA).[35] Both verbal and written information will be shared. Face-to-face meetings will be offered to eligible services. Indigenous project staff will be employed in recognition of their specialist skills and knowledge of working with ACCHSs. Additional time will be factored into project timelines for

skills and capacity to develop, refine and deliver systems for alcohol care. These supports will be based on current evidence and be in-line with services' cultural-care practices. Twenty-two ACCHSs will be recruited across Australia. Services will be randomised into two arms, an early support (intervention) and a late support (wait-control) arm (figure 2).

| Table 1 | Eight core support elements* | |
|---|---|---|
| 1 | Service champions | Services asked to nominate two representatives to act as advocates of alcohol care and links between the service and research team |
| 2 | National workshop | National capacity building workshop for service champions at the start of the support phase (and for early support services, a wrap-up feedback workshop at the end of the maintenance support phase) |
| 3 | Onsite training | Training will be offered to each ACCHS at their service |
| 4 | Resources/funding | Some resources (eg, visual resources for brief intervention and clinical guidelines) will be given to services for free. Additional funding will be provided for the selection and purchase of further resources (eg, FASD doll, standard drink cups). |
| 5 | Practice software support | Support will be provided to services throughout the trial to facilitate the routine clinical use of alcohol-related items in the practice software. |
| 6 | Data feedback | Individual service data will be fed back to each service every second month |
| 7 | Phone conferences | Phone conferences will be held every second month between service champions and project team (including an addiction medicine physician). These will allow sharing of ideas and joint problem solving. |
| 8 | Online platform | Further resources and information will be shared via a secure online platform. |

*These eight core elements of support will be tailored to local service needs. Further detail on each element is provided in the body of the text.
ACCHS, Aboriginal community controlled health service.

engaging with ACCHSs, including liaising with multiple key staff, follow-up conversations and knowledge translation of research documents.

### Inclusion criteria

Services eligible for the trial will meet the following criteria:
1. Are an ACCHS.
2. Provide care to 1000+clients per annum.
3. Use Communicare as their practice software.

### Exclusion criteria

Services will only be excluded if signed institutional consent forms are not returned before the close of recruitment. Data from clients aged 15 or under will not be extracted.

### Consent

After written institutional consent is signed, a memorandum of understanding will be negotiated by the senior investigator (KC) and lead author (KHH) with each ACCHS. Staff participating in individual qualitative interviews will also provide informed consent. Further consent will be sought, following completion of the main trial, if the authors wish to perform ancillary studies using stored data.

### Randomisation

ACCHSs will be categorised into three strata based on their remoteness using ASGC-RA[35] : (1) urban and inner regional; (2) outer regional and remote; (3) very remote. Service names will be replaced by consecutive numbers. Within each stratum, half the services will be randomised by a computer program into the early support arm and half into the late support arm. Randomisation will be conducted by an author (TD) who is blind to services' names.

### Sample size

Sample size requirements were calculated using PASS (Power Analysis and Sample Software).[36] Because larger numbers are needed to be able to determine increases in treatment provision than to demonstrate an increase in screening, the treatment provision outcome was used for sample size calculation. In an ACCHS of 1000+clients per year, approximately 60% (n=600) are likely to be aged 16 years or older.[37] We estimate that 57% of clients (n=342) have likely been screened for alcohol use within 12 months[16] and at least 25% (n=86) of screened clients are likely to be drinking above NHMRC recommended limits.[14 38 39] In the late support arm, it is likely that 60% (n=51) of identified unhealthy drinkers will have an alcohol intervention recorded.[16] Assuming an intra-cluster correlation coefficient of 0.04,[40 41] enrolling 10 early support services and 10 late support services will allow for an increase in treatment provision over a 12-month period of at least 13% in the early support services to be detected (ie, from 60% to 73%; 80% power and 2-sided significance of 0.05). We will enrol an additional service

in each arm to allow for the possibility of clusters dropping out of the study. Accordingly, the target number of services to be enrolled is 22.

### The model of support

ACCHSs will be acknowledged as the experts in cultural healthcare for Indigenous peoples in their communities and Australia. The study also recognises the need for flexibility based on needs of individual services and of the ACCHS sector as a specialist sector overall. Support approaches will include the same core elements (table 1). However, approaches will be tailored to the needs of each ACCHS, for what works best for their service, clients and communities.

The core support elements were designed based on evidence-based approaches for supporting implementation of alcohol care.[21 29] The project team also drew on their experience working within (six authors) and with Aboriginal health services; and in health workforce development (nine authors). Advice was also received from ACCHS peak organisations and networks in New South Wales and South Australia, and from a research team conducting a quality improvement trial for ACCHSs on diabetes screening.[42]

At the start of the early support phase, the support will be refined based on preliminary analysis of qualitative data from staff interviews and after feedback from the initial national workshop for early support service representatives. Tailoring and further minor refinement of support will occur during the trial, informed by service feedback.

Early support ('intervention') services will receive the support soon after the trial starts (1 year of active support, 1 year maintenance phase). Late support (wait-control) services will receive the same intervention elements 2 years after the start of the early-support phase. Due to the nature of the intervention, it is not possible to blind participants or staff as to which services are receiving support.

1. Identifying service champions
   Each service will be asked to identify two staff representatives to work collaboratively with their service staff and the research team on increasing alcohol care. Services will be asked to consider nominating a clinician, Aboriginal health professional and/or an individual from management. Service champions will be invited to attend an initial national face-to-face workshop, and second monthly teleconferences. These meetings will ensure that support approaches are in line with the values of each ACCHS, and will allow sharing of expertise and initiatives. Champions will be key in highlighting the potential benefits of change in their service and will encourage service staff to discuss their second monthly data feedback, and to address any key barriers to change.[43]
2. Workshops
   A national workshop will be held at the beginning of each support phase to bring champions from those

services together for capacity building and networking. Feedback will be obtained from champions about pre/post-workshop knowledge, training preferences, priority areas for support and useful resources. Key presentations from workshops will be video-recorded and made available via a password-protected website.

A wrap-up workshop will be organised for early support service champions (only) at the end of their support phase. Opportunities will be provided for service champions to provide verbal and written feedback on support, to discuss continuing quality improvement approaches and to network. Study results to that point will be discussed.

3. Onsite training

Training will be offered at each service for all staff, with maximum duration of 2 days. Core elements will include screening, alcohol and the body, brief intervention and treatment approaches, including relapse prevention medicines. Due to the individuality of each service, flexibility will be given to allow inclusion of other (alcohol-related) training topics. The training will be co-facilitated by an addiction specialist (KC) and an Aboriginal researcher with clinical background (KHH) and will use culturally specific content and resources. Training content will be aligned with Indigenous protocols such as gender appropriateness, kinship systems and cultural obligations; such a family and community sharing which at times can impact on substance use.[27 44] Pre/post-training feedback will be collected from participants.

4. Resources/funding

The trial will provide several hard copy and electronic resources for free to services (ie, the Handbook for Aboriginal Alcohol and Drug Work,[45] Alcohol Awareness Kit[46] and Quick Reference Guide to the Treatment of Alcohol Problems[47]).

Each ACCHS will be provided with funding to purchase additional resources. Services can choose, in partnership with the chief investigators, how they spend their funding. The early support arm of the project will receive up to $9000 for resources. The late support arm will receive up to $3500. This is a reduced sum as they will have access to resources developed within the early support phase, so time and development costs will be less. Resources must be alcohol-related and may include additional training, conference attendance, education resources for staff or clients, funds for health promotion events or for local adaptation of resources.

5. Practice software support

A Communicare support officer will be available during the trial to support ACCHSs with the practice software. Some ACCHSs may choose to make modifications to Communicare to make alcohol 'clinical items' more accessible for staff. One example is adding AUDIT-C into the Aboriginal and Torres Strait Islander (Indigenous) health assessment ('Adult Health Check'), which was not routine in standard versions of Communicare at the time of trial commencement. AUDIT-C can also be

1. Are you meeting your AUDIT-C screening goal?

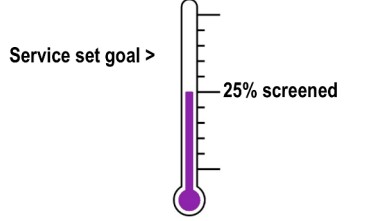

2. How many clients your service is screening

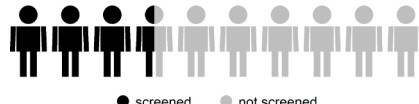

Screening rate over 12 months (out of clients aged 16+ who have attended in the past year)

3. How is your AUDIT-C screening rate going over time?

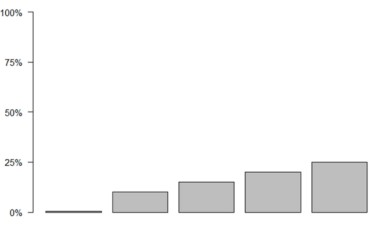

Screening rate for each 2-month period

**Figure 3** Three of the elements used to feedback alcohol screening data to ACCHSs.

added to other regular assessments such as antenatal and the pre-consult examination (ie, routine observations and/screening by a nurse or Aboriginal health worker before seeing the doctor). Services will also be provided with information about existing fields ('clinical items') available on Communicare for recording episodes of alcohol care (education/advice; on-site counselling), which can then be retrieved by data extraction.

6. Second monthly data feedback

Individual services will receive feedback on their routinely collected data on alcohol care every 2 months during their support phase. Data will be represented visually using images and or graphs (figure 3) and be fed back to nominated staff, including champions. Services will be asked for feedback on the data representation early in the support phase and adaptions will be made as requested. Services will be encouraged to discuss the results among staff, to see if there are ways to further improve alcohol screening and treatment rates.

7. Phone conferences

Second monthly phone conferences will be organised for service champions. These will provide the

opportunity for ACCHSs to network nationally. The champions will be split into two groups for the phone conferences, based on remoteness. During teleconferences, champions can discuss their data feedback, progress, successes, difficulties, helpful tools, resources or approaches.

8. Online platform

A password-protected web portal will allow ACCHS staff to access and share current evidence, culturally appropriate resources and to network. It will include a discussion board. Short videos, training presentation slides and clinical protocols will be uploaded by project staff.

### Data collection
#### Qualitative data

Qualitative data will be collected from ACCHSs at the start of the early support arm. The data will help inform the support model. Structured phone interviews will be conducted with approximately five staff members from each ACCHS in the early support arm, with the aim of a total of 50. Champions and/or key staff will be approached to assist with recruiting interviewees.

Alcohol can be sensitive to discuss due to cultural protocols such as kinship/family systems, gender or age, shame (embarrassment) or community and social acceptance of alcohol.[48–50] Therefore, the interview questions will be framed using a strengths-based, capacity-building approach.[51] Interview questions will include approaches to make it easier to talk about alcohol, treatment around alcohol, skills and knowledge needed to provide quality

alcohol care, new ways of helping people who struggle with alcohol, other successful programmes that could be used as a model for alcohol care and ideas for improvement in alcohol care.

Interviews will be conducted by two Indigenous staff with experience working with ACCHS and conducting interviews. One interviewer will facilitate the interview while the other will live-transcribe. The interviewers will compare and discuss notes after each interview. Memos will also be written by the lead author (KHH) on each interview.

#### Quantitative data

Non-identifiable data on episodes of screening, alcohol care and related variables (see below) will be extracted from the Communicare database of each ACCHS on the 28th day of every second month throughout the 5-year trial (figure 4). The baseline extraction will include data for 12 months before the trial starts. Participating services use structured query language (SQL) queries developed for this project to extract routinely collected data. For every 2-month period a denominator sheet will be extracted which includes the date of the last visit for each client who has attended during that period and their gender and age. In addition the following variables will be extracted, each one linked to an individual client ID and the date the variable was recorded: AUDIT-C responses; the clinicians' perception of the client's drinking status ('Ex-drinker', 'Non-drinker', 'Within safe drinking limits' or 'Unsafe - needs intervention'); whether a brief intervention, or a counselling session for alcohol use was provided; whether a relapse prevention medication was

| | Enrolment | Allocation | Post-allocation | | | | |
|---|---|---|---|---|---|---|---|
| **TIMEPOINT** | $t_{-1 (Feb 16)}$ Jul 2016 to Jun 2017 | 16/06/17 | $t_0$ 29/8/17 Year 1 | $t_1$ 28/8/18 Year 2 | $t_2$ 14/8/19 Year 3 | $t_3$ 13/8/20 Year 4 | Close out 28/4/21 |
| **ENROLMENT:** | | | | | | | |
| Eligibility screen | X | | | | | | |
| Informed consent | X | | | | | | |
| Allocation | | X | | | | | |
| **INTERVENTIONS:** | | | | | | | |
| Early support services | | | α | β. | | | |
| Late support services (wait-list control) | | | | | α | β | |
| **ASSESSMENTS:** | | | | | | | |
| **Baseline/outcome variables** Retrospectively collected for 2-monthly periods† from Feb 2016 to Feb 2021: AUDIT-C screening rate; provision of brief interventions‡, counselling; prescriptions for acamprosate, naltrexone, disulfiram; AUDIT-C results; systolic BP; GGT; HbA1c§ | X | X | X | X | X | X | X |

*Data will be retrospectively extracted for the 12-month period before the start of the support (the intervention). All data used in this study are routinely collected clinical data, extracted from the practice software ('Communicare'). †Data will be extracted on the 28th day of every second month. The first census date ($t_1$) is one year after the start of the intervention; $t_{2-4}$ are other census points. ‡Unhealthy alcohol use = hazardous consumption or alcohol use disorders §Blood test results (GGT, HbA1c) are only available when these have been routinely conducted. αActive support phase βMaintenance support phase

**Figure 4** Project timeline: SPIRIT schedule of enrolment, interventions and assessments for the trial.

prescribed (ie, acamprosate, naltrexone or disulfiram); and if measured, a range of biological markers which may be affected by alcohol consumption (BP, HbA1C and GGT).

A spreadsheet will track dates when key elements of support are provided to each service. Staff numbers attending training and participating in teleconferences will be recorded.

### Service retention and withdrawal

Regular contact will be made with services during their support phase through the second monthly teleconferences (by KC and often KHH), and for the purposes of data extraction (by BH). Effort will be made to address any concerns or suggestions, and to minimise burden of participation on services. However, services may withdraw from the study, for any reason and at any time.

### Data storage and confidentiality

De-identified data will be sent to the project team every 2 months (by email or Cloudstor). Data will be stored securely in a University of Sydney drive only accessible by relevant members of the research team. Identification numbers rather than names are used for services in the data set and in reporting. Data will stored for 7 years after the last publication on the study.

### Analysis
#### Qualitative

Two independent analyses of the qualitative data will be conducted: analysis of the memos and of the transcripts. This complementary analytical strategy aims to enhance examination of the raw data in the formation of key concepts.[52]

Interview transcripts and memos will each be imported into the qualitative software, NVivo V.11. Memos will be analysed by the lead interviewer (KHH). Transcripts from the structured interviews will be analysed by an independent research associate to produce key themes. This involves the interpretation of descriptive data to explain the meanings of the interviewees' responses.[53] Each data set will be coded separately using a process of constant comparison.[53] This will involve coding and organising like data into categories along with a comparison and interpretation of emerging themes across all transcripts and memos.[54] A team of relevant experts will meet to discuss both analyses and find consensus on key themes and sub-themes. These key themes and findings will be described.

#### Quantitative

Only the routinely collected data on Indigenous Australian clients aged 16+ years who have attended a participating ACCHS in the last 12 months will be included in analysis to assess:

1. The odds of a client being screened using AUDIT-C at least once in any 2-month period (considering those who attend in that period).
2. The odds of a client being recorded as receiving treatment for unhealthy alcohol use at the ACCHS, includ-

ing advice/education or counselling, relapse prevention medicines.
3. The number of clients identified by ACCHS staff to have unhealthy alcohol use:
   a. via AUDIT-C: using cut-offs recommended for ACCHSs by the Australian Institute of Health and Welfare (3+ for a woman, 4+ for a man).[23]
   b. and/or staff perception of drinking status (eg, safe, unsafe, non-drinker or ex-drinker).

We will use statistical packages, IBM SPSS for descriptive analyses and R for multi-level modelling.

Multi-level modelling will be used to account for clustering of observations under clients, who are nested within services. We will assess if clients are more likely to be screened for risky alcohol-use and provided with treatment if they are attending a service receiving support.

Two dummy variables will be constructed to indicate whether or not clients were screened with the AUDIT-C during each 2-month data extraction period, or if they received treatment. These variables will be used as outcomes in multi-level logistic regressions. As data are aggregated at bimonthly intervals, repetitive screenings within those 2-monthly periods will not influence the modelled effect of the intervention. However, to ensure that the intervention is associated with increased odds of appropriate screening, we will create a dummy variable indicating whether a client has been screened in the previous 12 months, and use this variable as the outcome in a multi-level logistic regression.

Characteristics of ACCHSs, such as number of clients, will be compared descriptively between the early/late-support services. Characteristics that display major imbalance between arms will be considered as adjustment factors in outcome analyses. As data are collected as part of routine clinical activity, missing data are likely to be non-random, as such multiple imputation will not be performed.

All analyses will be repeated for the maintenance phase of the study to see if study effects weaken or strengthen over time. The analysis will also be repeated after the late support services have received support, to see if any effect on use of screening or treatment can be replicated.

If a significant effect of the support on alcohol care outcomes is demonstrated, a secondary analysis will examine the elements of support which were most effective, and whether the effect of support changed over time during a support phase. Potential change in client health indicators over time (AUDIT-C scores, GGT, HbA1C, BP) will be examined in relation to support provision.

### Ethics approvals

Ethics approval was obtained by eight ethics review committees, including three Aboriginal-specific committees. As there are minimal foreseeable risks in this study, there will not be an independent committee monitoring trial conduct, data or adverse events. Feedback from services will be provided to investigators and adaptations made as necessary.

**Table 2** WHO trial registration data

| Data category | Information |
|---|---|
| Primary registry and trial identification number | Australia New Zealand Clinical Trials Registry: ACTRN12618001892202 |
| Date of registration in primary registry | 21/11/2018 |
| Secondary identifying numbers | APP1105339 |
| Sources of monetary or material support | NHMRC, Australia |
| Primary sponsor | The University of Sydney |
| Secondary sponsor | Royal Prince Alfred Hospital |
| Contact for public and scientific queries | Kate Conigrave, MBBS, FAChAM, FAFPHM, PhD kate.conigrave@sydney.edu.au |
| Public title | Supporting Aboriginal community controlled health services to deliver alcohol care: a cluster randomised controlled trial |
| Scientific title | Increasing uptake of evidence-based management of unhealthy alcohol use in Aboriginal primary healthcare services: a cluster randomised controlled trial |
| Counties of recruitment | Australia |
| Health condition(s) or problem(s) studied | Hazardous alcohol use; alcohol use disorders |
| Interventions | Health service support (including training, sharing learning between services); regular data feedback |
| Key inclusion or exclusion criteria | Health services:<br>1. are Aboriginal Community Controlled Health Services<br>2. deliver care to at least 1000 unique clients annually<br>3. use 'Communicare' as their practice software<br>Data from clients 16 years and older are eligible for extraction |
| Study type | Interventional;<br>Allocation: randomised;<br>Primary purpose: prevention;<br>Cluster randomised trial |
| Date of first enrolment | 28/10/2016 |
| Target sample size | 22 services |
| Recruitment status | Complete |
| Primary outcomes | AUDIT-C screening rate |
| Key secondary outcomes | Brief intervention for alcohol rate |

AUDIT-C, Alcohol Use Disorders Identification Test-Consumption; NHMRC, National Health and Medical Research Council.

## Patient and public involvement in the research

This trial was suggested by the Aboriginal Health Council of South Australia, the peak organisation for South Australian ACCHSs. Service participation in the trial must be approved by the board of each local ACCHS. The board is recognised as representing patients and community in such decisions. As described earlier, service staff will be consulted throughout their active support phase on how project support can best be tailored. Results will be fed back bimonthly to services in an accessible manner for all staff and board members, and services will have opportunity to provide comment on draft outcome reports before publication.

## Dissemination of findings

After feedback to participating ACCHSs, results will be disseminated via peer-reviewed publications and conference presentations. Publications will be led by study investigators, research students or staff supported by the investigators. The resources developed through this trial will also be made freely available electronically at the end of the study period to support other health services.

## Data sharing

It is not possible to make the data set publicly available because of ethical constraints. Alcohol is a sensitive issue, and the data belong jointly to the 22 services who take part. For the main outcome report, statistical code used to analyse data will be made publicly available.

## Authorship policy

All grant holders, and those involved in study conception and design, will co-author the main outcome piece. Further articles based on sub-analyses or secondary research questions will be authored by those directly involved in those questions. Acknowledgement will be given to individuals involved in design of the main paper.

## Protocol amendments

The trial's registration data are shown in table 2. Any necessary protocol amendments will be agreed on by study investigators, and where appropriate, participating services. Then they will be registered with the Australian and New Zealand Clinical Trials Registry.

## DISCUSSION

To our knowledge, this is the first large-scale randomised trial testing whether external support can enhance uptake of evidence-based alcohol care in services for Indigenous peoples. This study will add knowledge in the field of screening, early intervention and treatment for Indigenous peoples of Australia and inform future research and policy development. The findings are also likely to be relevant for Indigenous peoples internationally who have similar experiences of colonisation and inter-generational trauma. The research will also have relevance for non-Indigenous health services.

A strength of this study is its feasibility across a large number of services due to the use of routinely collected data. However, it is not within the project's resources to assess the quality of data recording. For example, it is likely that some brief discussions of alcohol use will go unrecorded, or will be entered as free text (which is not readily extractable), rather than in the specified fields ('clinical items'). Also, some forms of alcohol risk are not reflected in the AUDIT-C, for example, drinking while pregnant. While we examine health indicators as secondary outcomes (AUDIT-C score, BP, HbA1C, GGT), these may not be sensitive or specific enough to be allowed a confident assessment of any reduction in alcohol-related risk or improvement in client health. AUDIT-C scores may in fact become higher with improved quality of screening. In future, data linkage studies (eg, examining hospital presentations) could allow more definitive assessment of health benefits of a model of service-wide support.

The model of support is designed to be culturally appropriate, collaborative, and tailored to individual service needs. The model is based on the use of champions within services and feedback of routinely collected data, so has potential to be sustainable, allowing for continuing quality improvement. The model also has potential to be scaled up for longer-term support of ACCHSs Australia-wide. The need for tailoring and flexibility in support elements provides challenges for analysis. However, this approach is ethically and practically necessary for working in partnership with Indigenous services. This type of local tailoring also has value for non-Indigenous clinical services in culturally and geographically diverse regions.

## Author affiliations

[1]Sydney School of Public Health, Faculty of Medicine and Health, University of Sydney, Camperdown, NSW, 2006
[2]Discipline of Addiction Medicine, Sydney Medical School, Faculty of Medicine and Health, The University of Sydney, Camperdown, New South Wales, Australia
[3]Centre for Alcohol Policy Research, La Trobe University, Melbourne, Victoria, Australia
[4]School of Public Health and Community Medicine, University of New South Wales, Sydney, New South Wales, Australia
[5]Aboriginal Drug and Alcohol Council of South Australia, Underdale, South Australia, Australia
[6]Southern Queensland Centre of Excellence in Aboriginal and Torres Strait Islander Primary Health Care (Inala Indigenous Health Service), Inala, Queensland, Australia
[7]School of Medicine, Griffith University, Nathan, Queensland, Australia
[8]Illawarra Aboriginal Medical Service, Wollongong, New South Wales, Australia
[9]Drug Health Services, Sydney Local Health District, Camperdown, New South Wales, Australia
[10]Aboriginal Health Council of South Australia, Adelaide, South Australia, Australia
[11]National Drug Research Institute, Curtin University Bentley Campus, Perth, Western Australia, Australia

**Acknowledgements** This work was supported by the National Health and Medical Research Council funding through a Project Grant (#1105339); a Centre of Research Excellence in Indigenous Health and Alcohol (#1117198); and a Practitioner Fellowship for K Conigrave (#1117582). The funding body had no contribution to study design, writing of report or in the decision to submit this report for publication; nor will they influence the submission of future reports. Thanks to Dr David Scrimgeour and to the Aboriginal Health Council of South Australia for suggesting this study. Also to Professor Sandra Eades, Professor Robert Sanson-Fisher and Mr Paul Ishiguchi for advice on study design and management.

**Contributors** KHH refined the protocol for cultural appropriateness, drafted the protocol report, reviewed literature, recruited services, and will play a key role in training, qualitative data collection and qualitative analysis. KC, PSH, NH, RI and KSKL conceived the study design. SW and NH assisted with ensuring design was culturally appropriate; KC, PSH and RI provide clinical expertise; BH and DJ provided guidance on Communicare use, data extraction and ACCHS support; BH assisted with service recruitment leads data extraction and facilitates service liaison. TD developed the analysis plan, including power analyses, and provides ongoing advice. JHC is responsible for merging and curating the data, and conducting t1 quantitative analysis. KC, PSH, RI, DG, TD, KSKL, SW and DJ are grant holders. All grant holders are on the project steering committee; all authors contributed to the development of the study protocol and approved this manuscript.

**Funding** The National Health and Medical Research Council, Australia

**Competing interests** None declared.

**Patient consent for publication** Not required.

**Provenance and peer review** Not commissioned; externally peer reviewed.

**Data availability statement** No data are available.

**ORCID iD**
Katherine Conigrave http://orcid.org/0000-0002-6428-1441

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
