## [Reviewer comments · BMJ Open]

ARTICLE DETAILS

TITLE (PROVISIONAL)	Supporting Aboriginal Community Controlled Health Services to deliver alcohol care: protocol for a cluster randomised controlled trial
AUTHORS	Harrison, Kristie; Lee, KS; Dobbins, Timothy; Wilson, Scott; Hayman, Noel; Ivers, Rowena; Haber, Paul; Conigrave, James; Johnson, David; Hummerston, Beth; Gray, Dennis; Conigrave, Katherine

VERSION 1 – REVIEW

REVIEWER	Andreas Kimergård King's College London, UK
REVIEW RETURNED	29-Apr-2019

GENERAL COMMENTS	This is a very interesting RCT with great potential to improve alcohol care for indigenous populations. Thanks for the opportunity to review the manuscript. As this is a trial protocol (possibly of an ongoing trial), making changes can be difficult. The issues raised below can either be implemented in the manuscript if possible, or used as discussion and reflection points (why the authors chose differently) in the manuscript. I have one overall comment regarding the trial design and minor comments addressed in more detail below. As this trial intends to recruit services/participants whilst continuously developing and possibly improving the quality of the intervention how might this impact on the strategy for data analysis? A likely scenario is that the intervention will gain effectiveness over the course of the trial and so how would the data collected in the early stages compare to data collected in the later stages where more work have gone into intervention development. Another approach in trial design, where studies develop and adopt new and innovative interventions, would be to have a test phase/pilot trial including intervention development, acceptability to the study population and feasibility (can it be implemented in the recruited services) before proceeding to a full-scale trial. Here, the concern is the variation in the intervention over time combined with the apparent lack of feasibility and acceptability data (will participants accept/take part in the intervention?). It would strengthen the trial protocol if the authors could reflect on these challenges and account for the strengths of their design in greater detail. Other comments: Abstract
---

	Line 8: It would be useful to present a definition of the term 'unhealthy'. See also comments below regarding the use of the term. Line 14: What might these competing priorities be? Line 20: Please define 'external support'. It would improve the quality of the abstract if a clear description of the intervention was provided under the heading 'Intervention'. Line 25: Which type of services? Perhaps include under a new heading 'Setting'. Line 33: Are project champions health professionals or service users? Manuscript Page 3: Aims: Define 'remote support'. How will the study achieve its secondary aim (evaluate a service-wide model)? Page 3: Please provide further information about the 'quality research' briefly introduced on Page 3 and the qualitative interview component introduced one page 4. How do they each link up with, and contribute to, the study design? Page 3: Please include a sentence to say that the effectiveness of the intervention will be measured against care-as-usual. Page 6: Row 4: what type of resources? Page 12: Line 44. How will this be measured and validated? General comments: Terminology: I would recommend that the term 'unhealthy' drinking is defined in the beginning of the manuscript as this does offer some ambiguity and potential for variation in interpretation amongst readers. There is also some inconsistency between the use of this term and the included literature, e.g. reference 10 which investigated alcohol 'misuse', not unhealthy drinking as defined in the manuscript. Please also note that the terms safe/unsafe are used on page 10. Are they different from healthy/unhealthy? Consistent use of terms such as harmful, hazardous and AUD (one page 1) might provide further clarity and offer the opportunity to screen participants with validated screening tools. On page 10, will the health professionals be using a validate screening tool to determine the risk of drinking (e.g. AUDIT). If not, how will the trial compare 'perceptions by health professionals' (line 59-60) across different sites. A concern might be that different health professionals will perceive unhealthy drinking differently, e.g. according to training, experience, knowledge of clients and context. It would be interesting to add under Discussion if findings from this trial will be used to measure effectiveness of screening on treatment outcomes (e.g. alcohol consumption, health, hospital admissions). Is it assumed that an increased rate of screening (if
--	---

	this is the outcome of the intervention) will result in reduced drinking/better health? PPI: Will service users be able to influence and comment on the acceptability of the intervention? Have the authors done any work to assess acceptability and feasibility of the intervention prior to the trial commencing? If so, please include details in the manuscript/reasons not to.
--	---

REVIEWER	Martyn Symons National Health and Medical Council FASD Research Australi Centre of Research Excellence, Telethon Kids Institute, University of Western Australia, Australia
REVIEW RETURNED	14-May-2019

GENERAL COMMENTS	Overall, this seems like a great study and I believe it will add valuable evidence to the field. I think that the protocol would be strengthened by giving some more detail in some sections. 1. Is the research question or study objective clearly defined? A key part of the aims is to determine if “remote support” can result in increased uptake. However, there is nothing mentioned in the Introduction about remote support so it is difficult to determine if it is likely that it will help. I would recommend at least a few references from other studies that have taken this approach. 2. Is the abstract accurate, balanced and complete? Would be good to mention in the abstract that services will be stratified by remoteness before randomisation. 3. Is the study design appropriate to answer the research question? In terms of the overall design, there are a few things that are not clear to me. I think clarifying these would help the reader understand the design. 1. Why are there two years until the wait-list control is provided with the intervention? Is it expected to take two years before the intervention is fully implemented, or has the desired effects? The intervention itself doesn’t seem like it will take that long to roll-out. If the reasoning here could be explained in more detail that would be beneficial for the reader. If it is a matter of continuous refinement over the period, then would results be expected to be seen more quickly in the wait-list group once the intervention is more developed? 2. Is the bimonthly data collection primarily for feedback purposes to help support the intervention? Or is this to get more fine-grained detail of change over time? I believe that there are a wide range of things that could potentially affect HbA1C and systolic blood pressure measurements over five years more than alcohol dependence. Eg. Onset of diabetes, gestational diabetes, change in diet or exercise. To make a case that these will be effective measures of treatment uptake and improvement in consumption I would need to have more evidence presented in the Introduction. Will all people have regular GGT and HbA1c measures?
--

In research I have been involved in with Communicare some of this type of data is recorded in text fields or patient notes and not always in the correct field in Communicare. To check data entry quality it would be nice to see a record review of all notes/data collected on the patient of a percentage of files to ensure that data entry compliance is good.

When calculating sample size, it would be nice to have the final numbers used based on the rates provided for age, screening, and unhealthy drinking. Perhaps an indicative count something like below? What software was used for the calculation?

Condition	Per Site
Total clients seen	1000
Correct age (60%)	600
Screened for alcohol use (57%)	342
Second stage intervention recorded (60%)	205

4. Are the methods described sufficiently to allow the study to be repeated?

Some allowances must be made for the fact that interventions will be tailored for each site by the site themselves to make them culturally congruent. Some additional methodological issues could be explained further.

It appears that researchers are from many different locations. Will anyone be accessing the data stored at the University of Sydney remotely to do any analysis? How will the data be transferred from the ACCHSs to the University securely? How long will the data be kept before being destroyed?

Is there an AUDIT-C cut-off used to determine if clients are drinking at unhealthy levels and should be recommended to further treatment? Could this process be made more explicit? How is it determined if brief intervention or counselling is indicated?

5. Are research ethics (e.g. participant consent, ethics approval) addressed appropriately?

Good job on getting all of those ethics applications done!

6. Are the outcomes clearly defined?

1. The number of clients who are screened using AUDIT-C
2. The number of clients identified with unhealthy alcohol use
3. The number of clients who are offered treatment, including advice/education or counselling, relapse prevention medicines

At the end of page 8 it states: "Secondary outcomes will include recorded delivery of brief intervention, counselling or prescribed medicines to reduce relapse in alcohol dependence." But is this the case? In the list of three key goals above, from the quantitative data analysis section from page 17 suggests that the offer of treatment is being measured. However, again on line 53 on page 19 it states "records of treatment provided". It would be good if this could be clarified.

In terms of collecting data about counselling, will this only collect data on counselling provided by the service the data is being

	collected from? I have seen a number of studies that include referral to treatment but do not follow up on whether or not the subjects obtained treatment from any place which weakened the findings of the studies. Are there any service level variables that you will measure? These could potentially affect uptake/implementation of the intervention. E.g. number of staff, experience of the staff, self-efficacy of the staff, willingness of the staff to participate? 7. If statistics are used are they appropriate and described fully? The qualitative data analysis sounds solid. I have some questions about the qualitative analysis. On page 16 line 19 it states that "A spreadsheet will track dates when key elements of support are provided to each service". Will these be included in data analysis, given that data will be collected every 2 months? In the multi-level logistic regression, will clients be considered in terms of the number of screening episodes that they receive as it talks about "rates of screening". For the rates of screening, is it expected that everyone will be screened on every visit? Will you account for people who have multiple close successive appointments and expect that they will be screened every time (ie. If someone comes in for a second appointment two days later, it might be expected that they not get screened again?) Are you considering measuring pregnancy? Screening is even more vital during pregnancy and may affect rates of screening. Will you perform any statistical comparisons between active support and maintenance periods? Might they be expected to differ? 8. Are the references up-to-date and appropriate? As described in previous sections, I think that the Introduction could include a bit more information on particular aspects of the study. Mainly, is there evidence for this type of external support for staff and what are key components for success, and what is the evidence that getting treatment will affect the secondary variables more than other factors over the long time period. An update to paper 10 (2008) if available, or a systematic review of the area might be good. 13. Is the supplementary reporting complete (e.g. trial registration; funding details; CONSORT, STROBE or PRISMA checklist)? Yes, it appears so. 14. To the best of your knowledge is the paper free from concerns over publication ethics (e.g. plagiarism, redundant publication, undeclared conflicts of interest)? Yes. 15. Is the standard of written English acceptable for publication? Yes. I would recommend considering bimonthly, or every two months, instead of two-monthly but that is just my opinion.
--	--

VERSION 1 – AUTHOR RESPONSE

Response to Reviewer #1

COMMENT	RESPONSE
1. I have one overall comment regarding the trial design and minor comments addressed in more detail below. As this trial intends to recruit services/participants whilst continuously developing and possibly improving the quality of the intervention how might this impact on the strategy for data analysis? A likely scenario is that the intervention will gain effectiveness over the course of the trial and so how would the data collected in the early stages compare to data collected in the later stages where more work have gone into intervention development. Another approach in trial design, where studies develop and adopt new and innovative interventions, would be to have a test phase/pilot trial including intervention development, acceptability to the study population and feasibility (can it be implemented in the recruited services) before proceeding to a full-scale trial. Here, the concern is the variation in the intervention over time combined with the apparent lack of feasibility and acceptability data (will participants accept/take part in the intervention?). It would strengthen the trial protocol if the authors could reflect on these challenges and account for the strengths of their design in greater detail	We have now set out more clearly the process of developing and refining the core elements of the support provided before the support was provided individually to services. We have explained that subsequent improvements and tailoring, were more minor in nature. “The core support elements were designed based on evidence-based approaches for supporting implementation of alcohol care [21, 28]. The project team also drew on their experience working within (six authors) and with Aboriginal health services; and in health workforce development (nine authors). Advice was also received from ACCHS peak organisations and networks in NSW and SA, and from a research team conducting a quality improvement trial for ACCHSs on diabetes screening [40]. At the start of the early support phase, the support will be refined based on preliminary analysis of qualitative data from staff interviews (see below) and after feedback from the initial national workshop for all early support service representatives. Tailoring and further minor refinement of support will occur during the trial, informed by service feedback.” (Page 7, paragraphs 1 and 2) We have also added an explanation of how potential improvement of the intervention over time will be examined in secondary analyses: “a secondary analysis will examine the elements of support which were most effective, and whether the effect of support changed over time during a support phase.” (Page 14, paragraph 2) We have further clarified the strengths and limitations of this study design in a limitations paragraph in the discussion:

	“The need for tailoring and flexibility in support elements provides challenges for analysis. However, this approach is ethically and practically necessary for work in partnership with Indigenous services. This type of local tailoring also has value for other non-Indigenous clinical services in culturally and geographically diverse regions”. (see Page 19, paragraph 3)
2. Abstract: Line 8: It would be useful to present a definition of the term ‘unhealthy’. See also comments below regarding the use of the term.	The term ‘unhealthy’ has now been removed from the abstract due to the difficulty of providing a clear definition within abstract word limits. In the abstract we simply refer to alcohol-related harms (introduction), and alcohol screening (methods). (See also Reviewer 1, comment 12).
3. Abstract, Line 14: What might these competing priorities be?	We have now clarified within the abstract: “...there are many competing health, socio-economic and cultural client needs”
4. Abstract, Line 20: Please define ‘external support’. It would improve the quality of the abstract if a clear description of the intervention was provided under the heading ‘Intervention’.	We have removed the word ‘external’ as it appears to have caused confusion. We have clarified the description of support given and added a subheading, as suggested. We have used the subheading “The Support” rather than “Intervention”, as the term “intervention” can have very negative connotations for Aboriginal Australians, after a major and intrusive government initiative called the “Northern Territory Intervention”.
5. Abstract, Line 25: Which type of services? Perhaps include under a new heading ‘Setting’.	As suggested, we have added the subheading ‘Setting’ and have clarified that all services to be recruited in the study are Aboriginal community controlled health services (ACCHSs)
6. Abstract, Line 33: Are project champions health professionals or service users?	We have clarified that project champions are service staff: “nominate two staff as champions”
7. Manuscript: Page 3: Aims: Define ‘remote support’. How will the study achieve its secondary aim (evaluate a service-wide model)?	As per comment 1 above, we have deleted the term ‘remote’ as this has caused confusion (Page 3, paragraph 2). We have clarified the description of the model of support, so that it is more apparent that the support is directed at all staff of the service (not just doctors or nurses) and so is service-wide. (Pages 6-10 and Table 1 On considering the reviewer’s comment, we decided that this secondary aim (to ‘evaluate a service-wide model’) was redundant, as the primary gain of the study is about evaluating the service-wide model of support. (Page 3, paragraph 2)

8. Page 3: Please provide further information about the 'quality research' briefly introduced on Page 3 and the qualitative interview component introduced on page 4. How do they each link up with, and contribute to, the study design?	We have removed the term 'quality research' we hope that it is clear from our protocol that we have put in place rigorous methods, and also because it may have caused confusion with the later description of 'qualitative' research. (Page 4, paragraph 1). We have now clarified how the qualitative interviews are used to inform the refinement of the model of support, near the start of the early support phase. This is explained in the two relevant places: a) Description of the model of support "At the start of the early support phase, the support will be refined based on preliminary analysis of qualitative data from staff interviews" (Page 7, paragraph 2). b) Description of data collection: "Qualitative data will be collected from ACCHSs at the start of the early support arm. The data will help inform the support model" (Page 11, paragraph 1).
9. Page 3: Please include a sentence to say that the effectiveness of the intervention will be measured against care-as-usual.	As suggested, we have added the words "compared with care as usual" at the end of the sentence that provides an overview of methods (Page 3, Last paragraph).
10. Page 6, Row 4: what type of resources?	In Table 1 (page 6), we have added examples of the types of resources that will be provided for free to services, and that can be purchased by the services after agreement from the project team: "Some resources (e.g. visual resources for brief intervention and clinical guidelines), will be given to services for free. Additional funding will be provided for the selection and purchase of further resources (e.g. FASD doll, standard drink cups)." (See: Table 1) In the body of the text (Resources and Funding; page 9) we explain more fully the nature of the resources, for example, on resources which can be reimbursed: "Resources must be alcohol-related and may include additional training, conference attendance, resources for staff or clients, funds for health promotion events or for local adaptation

	of resources (e.g. translation into local language/s)” (Page 9, paragraph 3).
11. Page 12, Line 44. How will this be measured and validated?	As previously described, AUDIT-C scores will be extracted from routinely collected clinical data. We have clarified the cut-off score used for defining individuals as unhealthy drinkers: “...using cut-offs recommended for ACCHSs by the Australian Institute of Health and Welfare (3+ for a woman, 4+ for a man)” (Page 13, paragraph 2). We now explain in the limitations paragraph of the discussion that is beyond the scope of this study, to validate the quality of screening with AUDIT-C. “...it is not within the project’s resources to assess the quality of data recording. For example, it is likely that some brief discussions of alcohol use will go unrecorded, or will be entered as free text (which is not readily extractable), rather than in the specified fields (‘clinical items’)” (Page 19, paragraph 2).
12. General comments: Terminology: I would recommend that the term ‘unhealthy’ drinking is defined in the beginning of the manuscript as this does offer some ambiguity and potential for variation in interpretation amongst readers. There is also some inconsistency between the use of this term and the included literature, e.g. reference 10 which investigated alcohol ‘misuse’, not unhealthy drinking as defined in the manuscript. Please also note that the terms safe/unsafe are used on page 10. Are they different from healthy/unhealthy? Consistent use of terms such as harmful, hazardous and AUD (on page 1) might provide further clarity and offer the opportunity to screen participants with validated screening tools.	As suggested, we have moved the definition of the term ‘unhealthy drinking’ the point where the term is first used (the definition was previously later in that same paragraph). We have defined the term using ICD 10/11 compatible terms: “...unhealthy drinking (i.e. of hazardous, harmful or dependent alcohol use)” (Page 1, paragraph 2) Throughout the paper, the use of the terms, hazardous, harmful or dependent are in keeping with ICD-10 or ICD-11 definitions. ICD does not provide an umbrella term that neatly covers all three states, hence our use of the term ‘unhealthy’. This is a well-recognised term, and is used by the International Network on Brief Interventions for Alcohol and Other Drugs (INEBRIA) (which we have now cited). We believe that it is well suited to the primary care setting, where prevention of health harms is a priority. We prefer ‘unhealthy’ use it to the term ‘misuse’ which has been used with a variety of definitions and sometimes with negative

	connotations (e.g. an implication of deliberate overuse of alcohol). We do not use the term 'misuse' at all in the body of our manuscript at all. As the reviewer points out, it only appears in the title of one or more citations in the reference section. Because of the wide range of terms (and definitions of these) that past authors have used, it is not possible to find any term that is universally agreed on. The terms 'safe' and 'unsafe' use are used purely to describe the way that drinking status has been recorded by ACCHSs for several years. We have clarified this point (and the limitations of this term) in the introduction “Prior to 2017, the Australian government had asked ACCHS staff to record clients' alcohol use as 'safe' or 'unsafe', 'non-drinker' or 'ex-drinker' [22]. However health practitioners may or may not have known current drinking guidelines or have applied these to assess the risk” (Page 2, paragraph 1)
13. General comments: On page 10, will the health professionals be using a validated screening tool to determine the risk of drinking (e.g. AUDIT). If not, how will the trial compare 'perceptions by health professionals' (line 59-60) across different sites.. A concern might be that different health professionals will perceive unhealthy drinking differently, e.g. according to training, experience, knowledge of clients and context.	As described in Comment 11 (above), we have clarified that risky drinking is defined using AUDIT-C responses as recorded routinely by health staff, and using the cut-off scores that are routinely used in those services. As described in the previous comment (comment 12), we have clarified in the introduction that ACCHS service staff routinely record the client's drinking status, but that it is not clear what criteria they use to categorise clients. That same drinking status variable is mentioned in the data extraction (page 11, final paragraph). We believe that it is now clear that these two routinely recorded measures of drinking are compared in the quantitative analysis (Page 13, point 3). We have reordered this point to become point 3 of the analysis, so that the two primary outcome analyse come first.
14. General comments: It would be interesting to add under Discussion if findings from this trial will be used to measure effectiveness of screening on treatment outcomes (e.g. alcohol consumption, health, hospital admissions). Is it assumed that an increased rate of screening (if this is the outcome of	We have clarified in the secondary analyses, that these health indicators will be examined: “In addition individuals' changes in client health indicators (AUDIT-C scores, GGT, HbA1C, BP) over time will be examined” (Page 14, paragraph 4)

the intervention) will result in reduced drinking/better health?	It is not within the scope of the funding constraints of this study to examine hospital admissions. However we have mentioned this type of study as an area for future research in the discussion section : “While we examine health indicators as secondary outcomes (AUDIT-C score, BP, HbA1C, GGT), these may not be sensitive or specific enough to be allow a confident assessment of any reduction in alcohol-related risk or improvement in client health. AUDIT-C scores may in fact become higher with improved quality of screening. In future, data linkage studies (e.g. examining hospital presentations) could allow more definitive assessment of health benefits of a model of service-wide support.” (Page 19, paragraph 2)
15. General comments: PPI: Will service users be able to influence and comment on the acceptability of the intervention?	We had previously mentioned the opportunity for service staff to comment on the acceptability of the intervention (e.g. through feedback at the national training workshop, and bimonthly teleconferences). As per the responses to comment 1 (above) we have further clarified the input of services in designing and refining the model of support
16. General comments: Have the authors done any work to assess acceptability and feasibility of the intervention prior to the trial commencing? If so, please include details in the manuscript/reasons not to.	As described in comment 1 (above) we have clarified that the core elements of the support model were developed not only based on the literature, but on years of prior experience working with (and in some cases in) ACCHSs, including in health workforce development, and based on input from the peak ACCHS agencies in two Australian states (South Australia and New South Wales) and from a prior research project on quality improvement on diabetes in ACCHS. (Page 7, paragraphs 1 and 2)

Response to Reviewer # 2

COMMENT	RESPONSE
i) Is the research question or study objective clearly defined? A key part of the aims is to determine if “remote support” can result in increased uptake. However, there is nothing mentioned in the Introduction about remote support so it is difficult to determine if it is likely that it will	As mentioned in response to Reviewer 1, comment 7, the word ‘remote’ has been removed as it has caused confusion. Instead we have provided a clearer description of the type of support provided (Pages 6- 7). We have added a brief overview of studies that

help. I would recommend at least a few references from other studies that have taken this approach.	demonstrate effective models of support for primary care services to optimise implementation of evidence-based alcohol care (Page 2, paragraph 3).
ii) Is the abstract accurate, balanced and complete? Would be good to mention in the abstract that services will be stratified by remoteness before randomisation.	We have clarified in the abstract that services will be stratified by remoteness before randomisation
iii) Is the study design appropriate to answer the research question? In terms of the overall design, there are a few things that are not clear to me. I think clarifying these would help the reader understand the design:  1. Why are there two years until the wait-list control is provided with the intervention? Is it expected to take two years before the intervention is fully implemented, or has the desired effects? The intervention itself doesn't seem like it will take that long to roll-out. If the reasoning here could be explained in more detail that would be beneficial for the reader. If it is a matter of continuous refinement over the period, then would results be expected to be seen more quickly in the wait-list group once the intervention is more developed? 	We have clarified that after the first year of active support provision to 'early support' (intervention) services, there is a maintenance year, where only minimal support (data feedback and second monthly teleconferences) is provided. The wait-list control receives the intervention in the following year: "Early support ('intervention') services will receive the support soon after the trial starts (one year active support, one year maintenance phase). Late support (wait-control) services will receive the same intervention elements two years after the start of the early-support phase." (Page 8, paragraph 1). This one-year delay provides a buffer in case there are delays in support implementation in early support services. During that year preliminary data analysis is conducted to establish if the model of support was effective. If effective it would be offered largely unchanged to the wait-control services. We have now clarified that major refinements to the support model happen near the start of the early support phase before individual services are visited, and that later refinements and tailoring will be more minor (see Reviewer 1, comment1). As explained in the response to Reviewer 1, comment 1, we have now described that we will use the date of support provision in the quantitative analysis to examine whether support provided later in the support phase is more effective than support provided early in the project (Page 14, paragraph 3). This same approach can be used to see if support provided in the late-support phase is more

	effective than that of the early support phase.
2. a) Is the bimonthly data collection primarily for feedback purposes to help support the intervention? Or is this to get more fine-grained detail of change over time?	The bimonthly data collection is important for both purposes: for regular feedback to participating services, and to provide fine-grained detail of change over time. We have checked all references to the bimonthly data use, to be sure that this is clear. The quantitative analysis section has been slightly expanded to show how bimonthly data is used to provide fine-grained detail of changes in alcohol care over time “Two dummy variables will be constructed to indicate whether or not clients were screened with the AUDIT-C during each 2-month data extraction period, or if they received treatment. These variables will be used as outcomes in multi-level logistic regressions.” (Page 14, paragraph 1). Also see the previous comment, for how the bimonthly data collection can be used to see if the support provided improves over time. We have also set out that the same bimonthly data collection is used for feedback, which is a core part of the support provided. This data feedback supports services to reflect on barriers to implementation of evidence-based care and ways of overcoming these. (Page 8, paragraph 2).
b) I believe that there are a wide range of things that could potentially affect HbA1C and systolic blood pressure measurements over five years more than alcohol dependence. Eg. Onset of diabetes, gestational diabetes, change in diet or exercise. To make a case that these will be effective measures of treatment uptake and improvement in consumption I would need to have more evidence presented in the Introduction. Will all people have regular GGT and HbA1c measures?	We agree that there are a range of conditions which could potentially affect HbA1C and systolic BP (and GGT). Alcohol can influence each of these measures, and alcohol can also affect the ability of an individual to care for their unrelated health conditions. The word limits do not allow for a full discussion of these complex issues, so instead we have added a phrase in the introduction, to say that alcohol can affect the levels of these markers and added several relevant citations: “We will also describe changes in reported drinking and in biological measures that can be affected by drinking such as gamma glutamyltransferase (GGT, a liver enzyme), HbA1C and systolic blood pressure (BP)”. (Page 4, paragraph 2). In the discussion we have now pointed the limitations of these secondary outcome measures

	and the role of future data linkage studies to provide more definitive assessment of client health outcomes from such a service-wide model of support: “While we examine health indicators as secondary outcomes (AUDIT-C score, BP, HbA1C, GGT), these may not be sensitive or specific enough to be allow a confident assessment of any reduction in alcohol-related risk or improvement in client health. AUDIT-C scores may in fact become higher with improved quality of screening. In future, data linkage studies (e.g. examining hospital presentations) could allow more definitive assessment of health benefits of a model of service-wide support.” (Page 19, paragraph 2). We have clarified that GGT and HbA1C are only available if tested as part of routine care. (Page 12, para 1)										
c) In research I have been involved in with Communicare some of this type of data is recorded in text fields or patient notes and not always in the correct field in Communicare. To check data entry quality it would be nice to see a record review of all notes/data collected on the patient of a percentage of files to ensure that data entry compliance is good.	We agree that the reliance of data extraction from the correct field (or Communicare ‘item’) is a limitation. In the discussion section we have acknowledged this, and explained that is not within the resources or scope of this study to conduct a record review of free-text notes: “A strength of this study is its feasibility across a large number of services due to the use of routinely collected data. However, it is not within the project’s resources to assess the quality of data recording. For example, it is likely that some brief discussions of alcohol use will go unrecorded, or will be entered as free text (which is not readily extractable) rather than in the specified fields (‘clinical items’)” (Page 19, paragraph 2)										
3. When calculating sample size, it would be nice to have the final numbers used based on the rates provided for age, screening, and unhealthy drinking. Perhaps an indicative count something like below? What software was used for the calculation?    Condition Per Site     Total clients seen 1000   Correct age (60%) 600   Screened for alcohol use (57%) 342   Second stage intervention recorded (60%) 205   	Condition	Per Site	Total clients seen	1000	Correct age (60%)	600	Screened for alcohol use (57%)	342	Second stage intervention recorded (60%)	205	As suggested, we have now added numbers for sample size calculations (in addition to the percentages which were previously provided). We have also added information about the software package used for this calculation. (Page 5, paragraph 4).
Condition	Per Site										
Total clients seen	1000										
Correct age (60%)	600										
Screened for alcohol use (57%)	342										
Second stage intervention recorded (60%)	205										

4. Are the methods described sufficiently to allow the study to be repeated? a) Some allowances must be made for the fact that interventions will be tailored for each site by the site themselves to make them culturally congruent. Some additional methodological issues could be explained further.	As described above (Reviewer 1, Comments 7 & 10), we have added further information to clarify the core elements of the support provided (Pages 6-10 and Table 1).
b) It appears that researchers are from many different locations. Will anyone be accessing the data stored at the University of Sydney remotely to do any analysis? How will the data be transferred from the ACCHSs to the University securely? How long will the data be kept before being destroyed	The analysis is only being conducted by University of Sydney staff members. If those staff members are offsite, there is secure access to the data store via a virtual private network. Due to concerns about word limits we haven't added that information, but are happy to if the editor desires it. We have added a note to the data security section to explain file transfer from services: "De-identified data will be sent to the project team every two months (by email or Cloudstor). (Page 12, paragraph 4). We note that none of the eight ethics committees raised concerns about file transfer of the de-identified data. The data is difficult for a non-expert to interpret. For example it comes as 11 small Excel files which must be merged by linking with the unique client ID. Each AUDIT-C item is in a separate file, and shows the response category e.g. for AUDIT-2: '4 (7, 8 or 9)' The service name is not on the file. We have clarified that data will be stored for: "...seven years after the last publication relating to the project" (Page 12, paragraph 4).
c) Is there an AUDIT-C cut-off used to determine if clients are drinking at unhealthy levels and should be recommended to further treatment? Could this process be made more explicit? How is it determined if brief intervention or counselling is indicated?	As described at Reviewer 1, Comment 11, we have clarified that the standard AUDIT-C thresholds recommended for ACCHSs were used to determine which clients' drinking was unhealthy (and so warrant further treatment, such as either brief intervention, counselling or treatment) (Page 13, paragraph 2)
5. Are research ethics (e.g. participant consent, ethics approval) addressed appropriately? Good job on getting all of those ethics applications done!	Thankyou! It was hard work!

6. Are the outcomes clearly defined?  • The number of clients who are screened using AUDIT-C • The number of clients identified with unhealthy alcohol use • The number of clients who are offered treatment, including advice/education or counselling, relapse prevention medicines a) At the end of page 8 it states: “Secondary outcomes will include recorded delivery of brief intervention, counselling or prescribed medicines to reduce relapse in alcohol dependence.” But is this the case? In the list of three key goals above, from the quantitative data analysis section from page 17 suggests that the offer of treatment is being measured. However, again on line 53 on page 19 it states “records of treatment provided”. It would be good if this could be clarified.	We have now clarified our phrasing throughout the manuscript to make it clear that we are measuring recorded provision of treatment (including, brief intervention, counselling, or pharmacotherapies) rather than the offer of treatment For example: “The resultant study will be able to use routinely collected outcome data but this relies on the accurate recording of screening and alcohol care provided to clients.” (Page iv, paragraph 3)
b) In terms of collecting data about counselling, will this only collect data on counselling provided by the service the data is being collected from? I have seen a number of studies that include referral to treatment but do not follow up on whether or not the subjects obtained treatment from any place which weakened the findings of the studies.	We have clarified that we are only collected data on on-site counselling It is not within the scope of this study to collect data on counselling at other services (i.e. after referrals). Also, access to such external specialised services varies greatly between participating services.

c) Are there any service level variables that you will measure? These could potentially affect uptake/implementation of the intervention. E.g. number of staff, experience of the staff, self-efficacy of the staff, willingness of the staff to participate?	For each service we will calculate the number of clients attending the service in any two-month period (Page 14, paragraph 1). We found this to be an easier measure of service size, than measuring staff numbers, which change regularly. It is not within the scope or resources of this study to formally measure the level of experience or self-efficacy of staff. We do record number of staff participating in training events or teleconferences. “Staff numbers attending training and participating in teleconferences will be recorded.” (Page 12, paragraph 2).
7. If statistics are used are they appropriate and described fully? The qualitative data analysis sounds solid. I have some questions about the quantitative analysis: a) On page 16 line 19 it states that “A spreadsheet will track dates when key elements of support are provided to each service”. Will these be included in data analysis, given that data will be collected every 2 months?	As described in the responses to Reviewer 1, comment 1, we have now explained that date of provision of key elements of support will be used in analysis to see if support effectiveness improved over time during a support phase (Page 14, paragraph 2)
b) In the multi-level logistic regression, will clients be considered in terms of the number of screening episodes that they receive as it talks about “rates of screening”. For the rates of screening, is it expected that everyone will be screened on every visit? Will you account for people who have multiple close successive appointments and expect that they will be screened every time (i.e. If someone comes in for a second appointment two days later, it might be expected that they not get screened again?)	We have clarified that our outcomes will assess the odds of a person being screened at least once in a 2 month period. As data will be aggregated to 2-monthly periods additional screenings over that time period will not affect the modelled effect of the intervention. In addition to this, in response to your feedback, we have added the fact that we will perform a supplementary analysis to assess the odds of an attendee being screened with the AUDIT-C at least once in the past year (using multilevel logistic regression). In that way, the analysis will check ensure whether increases in the odds of being screened is due to increases in beneficial (at least once per year screening), rather than frequent (e.g. 6+ times per year) screening. “As data is aggregated at bi-monthly intervals, repetitive screenings within the two-monthly periods, will not influence the modelled effect of the intervention. We will also examine whether a client has been screened at least once in the previous 12 months (multilevel logistic regression.” (Page 14, paragraph 1) While we have not included this information in the manuscript because of word limit constraints, we

	also have the capacity to examine the odds of any one individual being screened more than twice in a 2-month period. This can screen for excessive servicing.
c) Are you considering measuring pregnancy? Screening is even more vital during pregnancy and may affect rates of screening.	We agree that recommendations for safe drinking are very different in pregnancy. Currently, we do not have ethical approval to extract data on pregnancy from the practice software. We have mentioned the lack of this information in the limitations section (and note that this is also a limitation of published AUDIT-C scoring) “...some forms of alcohol risk are not reflected in the AUDIT-C, for example, drinking while pregnant.” (Page 19, paragraph 2).
d) Will you perform any statistical comparisons between active support and maintenance periods? Might they be expected to differ?	We have clarified that analyses will be repeated at the end of the maintenance phase (2 years after the start of the early support arm) to see if study effects attenuate or strengthen: “All analyses will be repeated for the maintenance phase of the study to see if study effects attenuate or strengthen over time. (Page 14, paragraph 2).
8) Are the references up-to-date and appropriate? a) As described in previous sections, I think that the Introduction could include a bit more information on particular aspects of the study. Mainly, is there evidence for this type of external support for staff and what are key components for success, and what is the evidence that getting treatment will affect the secondary variables more than other factors over the long time period.	As mentioned at Reviewer 2, comment i, we have updated the introduction to include information on the effectiveness of support models with some similarity to our own and on key components for success. In the introduction we have now mentioned that the health indicators we are examining can be influenced by alcohol (as per Comment 2b). However, in the limitations paragraph in the discussion, we have added the point that these indicators may not be sensitive or specific enough to show improvements over the study period (as per Reviewer 1, comment 14).
b) .An update to paper 10 (2008) if available, or a systematic review of the area might be good.	We have updated this 2008 reference to a 2018 Cochrane review on the same topic (now reference #9).
13. Is supplementary reporting complete : Yes	N/A
14. Is paper free from concerns over publication ethics: Yes	N/A
15. Is the standard of written English acceptable for publication? Yes. I would recommend considering bimonthly, or every two months, instead of two-monthly but that is just my opinion.	We have changed the term second monthly to bimonthly as suggested throughout the paper.